# Antibiotic-induced changes in the microbiota disrupt redox dynamics in the gut

Aspen T Reese[1,2], Eugenia H Cho[3], Bruce Klitzman[4], Scott P Nichols[5], Natalie A Wisniewski[5], Max M Villa[2], Heather K Durand[2], Sharon Jiang[2], Firas S Midani[6], Sai N Nimmagadda[7], Thomas M O'Connell[8], Justin P Wright[1], Marc A Deshusses[9], Lawrence A David[2,6,7,10]*

[1]Department of Biology, Duke University, Durham, United States; [2]Department of Molecular Genetics and Microbiology, Duke University, Durham, United States; [3]Department of Bioengineering, University of Pennsylvania, Philadelphia, United States; [4]Department of Surgery, Duke University Medical Center, Durham, United States; [5]Profusa, Inc., South San Francisco, United States; [6]Program in Computational Biology and Bioinformatics, Duke University, Durham, United States; [7]Department of Biomedical Engineering, Duke University, Durham, United States; [8]Department of Otolaryngology-Head and Neck Surgery, Indiana University School of Medicine, Indianapolis, United States; [9]Department of Civil and Environmental Engineering, Duke University, Durham, United States; [10]Center for Genomic and Computational Biology, Duke University, Durham, United States

**Abstract** How host and microbial factors combine to structure gut microbial communities remains incompletely understood. Redox potential is an important environmental feature affected by both host and microbial actions. We assessed how antibiotics, which can impact host and microbial function, change redox state and how this contributes to post-antibiotic succession. We showed gut redox potential increased within hours of an antibiotic dose in mice. Host and microbial functioning changed under treatment, but shifts in redox potentials could be attributed specifically to bacterial suppression in a host-free ex vivo human gut microbiota model. Redox dynamics were linked to blooms of the bacterial family Enterobacteriaceae. Ecological succession to pre-treatment composition was associated with recovery of gut redox, but also required dispersal from unaffected gut communities. As bacterial competition for electron acceptors can be a key ecological factor structuring gut communities, these results support the potential for manipulating gut microbiota through managing bacterial respiration.
DOI: https://doi.org/10.7554/eLife.35987.001

*For correspondence:
lawrence.david@duke.edu

## Introduction

Mammalian gut microbial communities are likely to be structured by both host- and microbial-associated factors. Extensive research to date has focused on how host factors like diet (*David et al., 2014*; *Claesson et al., 2012*), genetics (*Goodrich et al., 2014*), geography (*De Filippo et al., 2010*; *Yatsunenko et al., 2012*; *Arumugam et al., 2011*), and immune state (*Hooper et al., 2012*) shape the gut microbiota. Yet, work in free-living microbial systems reveals that bacteria typically play active roles in shaping their own environment (*Shi and Norton, 2000*; *Gobbetti, 1998*; *Goddard, 2008*; *Osono, 2005*; *Rui et al., 2009*; *Gerbersdorf et al., 2009*). Identifying how these drivers interact is necessary both for a more complete understanding of the gut microbiota and for

**eLife digest** The gut is home to a large and diverse community of bacteria and other microbes, known as the gut microbiota. The makeup of this community is important for the health of both the host and its residents. For instance, many gut bacteria help to digest food or keep disease-causing bacteria in check. In return, the host provides them with nutrients. When this balance is disturbed, the host is exposed to risks such as infections. In particular, treatments with antibiotics that kill gut bacteria can lead to side effects like diarrhea, because the gut becomes recolonized with harmful bacteria including *Clostridium difficile* and *Salmonella*.

Reese et al. have now investigated what happens to the gut environment after antibiotic treatment and how the gut microbiota recovers. Mice treated with broad-spectrum antibiotics showed an increase in the "redox potential" of their gut environment. Redox potential captures a number of measures of the chemical makeup of an environment, and provides an estimate for how efficiently some bacteria in that environment can grow. Some of the change in redox potential came from the host's own immune system releasing chemicals as it reacted to the effects of the treatment. However, Reese et al. found that treating gut bacteria in an artificial gut – which has no immune system – also increased the redox potential. This experiment suggests that bacteria actively shape their chemical environment in the gut.

After the treatment, bacteria that thrive under high redox potentials, which include some disease-causing species, recovered first and fastest. This, in turn, helped to bring redox potential back to how it was before the treatment. Although the gut's chemical environment recovered, some bacterial species were wiped out by the antibiotic treatment. The microbiota only returned to its previous state when the treated mice were housed together with non-treated mice. This was expected because mice that live together commonly exchange microbes, for instance by eating each other's feces, and the treated mice received new species to replenish their microbiota.

These findings are important because they show that the chemical environment shapes and is shaped by the bacterial communities in the gut. Future research may investigate if altering redox potential in the gut could help to keep the microbiota healthier in infections and diseases of the digestive tract.

DOI: https://doi.org/10.7554/eLife.35987.002

developing rational interventions. If ecological forces prove important, then compositional changes will likely be the result of feedbacks between the microbes and their environment.

Redox potential, a metric of the environmental capacity for reducing chemical reactions (*i.e.*, those involving the gain of electrons) to occur, is a composite measurement of various factors that influence gut microbiota structure (*Cowley et al., 2015*; *Friedman et al., 2017*; *Dhall et al., 2014*). Much of our knowledge for how gut redox potentials are determined involves host-associated pathways (*Spees et al., 2013*; *Rivera-Chávez et al., 2016*). Passive diffusion of oxygen from the epithelium increases redox potential and stimulates growth of aerobic microbes (*Espey, 2013*; *Albenberg et al., 2014*). Secretion of redox-active immune molecules such as reactive oxygen species or nitrate is known to be a feature of inflammation that imposes oxidative stress on commensal microbes and can be exploited by select pathogens to colonize the intestine (*Winter et al., 2010*; *2013*; *Faber et al., 2016*; *Spees et al., 2013*; *Rivera-Chávez et al., 2016*; *Kelly et al., 2015*; *David et al., 2015*). Maintenance of redox homeostasis in host tissue can also have spillover effects on luminal redox state (*Circu and Aw, 2011*).

Yet, redox potential is likely to also be shaped by microbial metabolism. In free-living microbial communities, variation in available electron acceptors dictates where microbes that employ respiration can thrive; the differential microbial metabolism that follows can produce further changes in electron acceptor availability (*Morris and Schmidt, 2013*; *Chen et al., 2017*; *Noll et al., 2005*; *Orcutt et al., 2011*). Moreover, microbial metabolism has been proposed as a mechanism for low redox potential states in the lung of cystic fibrosis patients (*Cowley et al., 2015*).

Here, we investigated the nature of redox potential dynamics under antibiotic treatment to assess the importance of host and microbial processes in structuring gut bacterial communities. Antibiotics directly disturb the microbiota but are also expected to alter host biology related to redox potential.

Specifically, antibiotics have been found to increase gut epithelium oxygenation as a result of altered microbial composition and metabolic signaling to the host (*Kelly et al., 2015*; *Rivera-Chávez et al., 2016*). Increases in luminal oxygen due to diffusion from the microvasculature supplying the epithelium would lead to a higher redox potential under antibiotic treatment. In addition, host inflammation responses to antibiotic treatment and antibiotic-associated pathogen colonization have been shown to produce electron acceptors and other redox active molecules that cause oxidative stress (*Faber et al., 2016*; *Winter et al., 2010*; *2013*; *Spees et al., 2013*). While these pathways have been demonstrated previously, their overall impact on redox potential has not been measured. Furthermore, the contribution of microbial metabolism to gut redox potential has not been tested, although it is known that a wide range of resident gut bacteria can respire aerobically and anaerobically (*Ravcheev and Thiele, 2014*). Antibiotic-driven bacterial inhibition could increase the availability of electron acceptors and thus serve as another mechanism for antibiotic induced changes in gut redox potential. On the other hand, antibiotic inhibition could limit bacterial production of oxidizing agents thereby resulting in an overall decrease in redox potential under antibiotics.

Changes in redox potential under antibiotics would in turn be expected to yield insight into the forces structuring the composition and function of the microbiota. Elevated redox potential due to a host immune response could restrict the microbiota beyond direct antibiotic mortality as inflammation would introduce additional oxidative stress into microbial ecosystems. By contrast, elevated redox potential due to the accumulation of oxygen or anaerobic electron acceptors would foster the growth of respiring bacteria. Antibiotic disturbance produces reproducible community succession in the gut following treatment—most notably, a transient bloom in Enterobacteriaceae (*Antonopoulos et al., 2009*; *David et al., 2015*; *Young and Schmidt, 2004*; *Theriot et al., 2014*; *Peterfreund et al., 2012*; *Dethlefsen et al., 2008*; *Jakobsson et al., 2010*; *Looft and Allen, 2012*), which has been ascribed to an increase in oxygen availability (*David et al., 2015*). But, the role of redox potential during this successional process has not previously been studied. If redox potential changes due to antibiotic treatment, we would predict that redox state recovery would be necessary for community resilience, that is for compositional recovery to a pre-disturbance state to occur (*Shade et al., 2012*). Furthermore, we expect that feedback between the community (*i.e.* the biotic component of the system) and their environment (*i.e.*, the abiotic component) would drive further redox potential changes during the successional period. Such a pattern would highlight the potential of manipulating redox potential to alter community dynamics after disturbance.

Here, we combined in vivo and ex vivo antibiotic studies to isolate the effects of host and microbial pathways on redox state in the mammalian gut. We confirmed that redox potential increased during antibiotic treatment in association with some changes in host immune state. However, multiple lines of evidence in both mice and an artificial human gut model suggested that antibiotic-induced changes in microbial metabolism were sufficient to cause an increase in redox potential. After antibiotic treatments ended, we observed redox recovery within a week. A successional return to conventional community composition only occurred, though, when mice were co-housed and shared gut microbiota, indicating that microbial dispersal is necessary above and beyond environmental recovery for the return of normal microbiota community structure.

## Results

### Antibiotics caused a significant increase in mouse fecal redox potential within a day of treatment

For five days, we orally gavaged a cohort of conventional mice with a cocktail of antibiotics (ampicillin, vancomycin, metronidazole, and neomycin [*Reikvam et al., 2011*]) to broadly inhibit gut bacteria. We measured redox potential in freshly voided feces with a microelectrode paired with a reference electrode daily. Within sixteen hours after the first dose of antibiotics, redox potential significantly increased from $37 \pm 164$ mV at baseline to $227 \pm 45$ mV (p=0.04 Bonferroni-corrected Mann-Whitney U test; *Figure 1A*). Throughout treatment, redox potential differed overall between treated and control mice (p=0.005 linear mixed effects model likelihood tests).

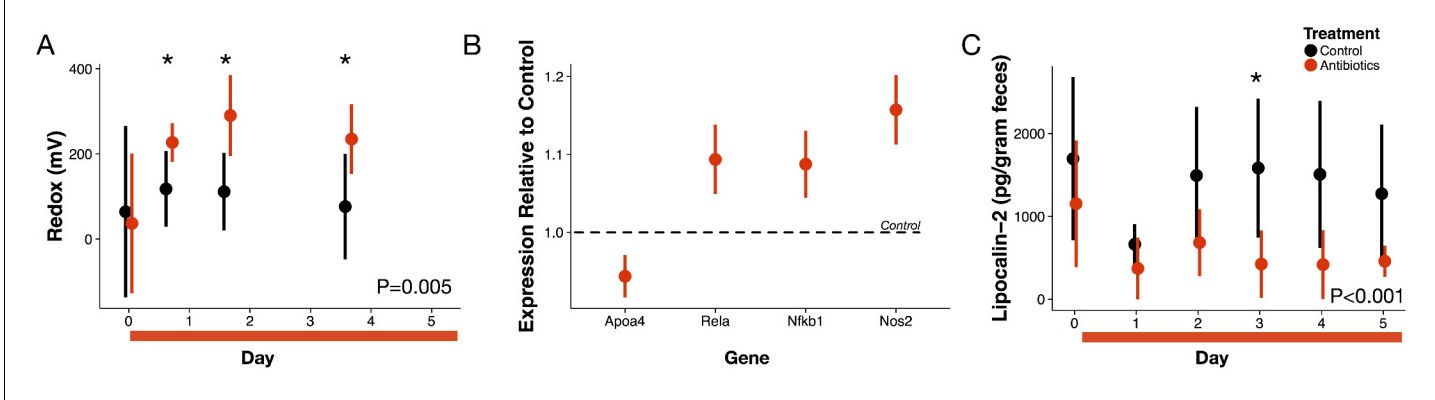

**Figure 1.** Antibiotic treatment effects on gut redox state and host inflammation. (A) Redox state measured in freshly voided feces of treated (red) and control (black) mice (n = 9–10 per treatment) differed (p=0.005, linear mixed effects model likelihood tests). Replicate data is presented in *Figure 1—figure supplement 2*. (B) Gene expression for four inflammation associated genes measured with RT-qPCR of RNA isolated from feces on the final day of antibiotic treatment (n = 6–7) differed from control levels (p<0.05, Bonferroni-corrected one-sample *t*-tests). (C) Intestinal inflammation measured as fecal concentration of the biomarker lipocalin-2 (n = 9–10 per treatment) differed between treated and control mice (p<0.001 linear mixed effects model likelihood tests). Data are shown as means ± SD. Post-hoc test results for individual time points (* indicates p<0.05 Bonferroni-corrected Mann-Whitney U test) are included for visualization purposes. Antibiotic treatment began after Day 0 measurement; red bars indicate treatment duration.
DOI: https://doi.org/10.7554/eLife.35987.003

The following figure supplements are available for figure 1:

**Figure supplement 1.** Overall antibiotic effects on inflammation.
DOI: https://doi.org/10.7554/eLife.35987.004

**Figure supplement 2.** Redox potential data during treatment from replicate experimental run.
DOI: https://doi.org/10.7554/eLife.35987.005

## Evidence antibiotics alter host biology to increase redox potential

We examined first whether antibiotics affected redox potential through direct host effects mediated by an immune response. Expression of three host genes in fecal samples were consistent with the hypothesis that redox shifts were associated with immune activation. We measured increases relative to controls in *Nos2*, which is linked to reactive nitrogen species levels (*Dedon and Tannenbaum, 2004*; *Winterbourn, 2008*); increases relative to controls in *Rela*, which has been found to contribute to IBD type inflammation in mice via a proinflammatory cytokine response (*Waddell et al., 2013*); and, decreases relative to controls in *Apoa4*, which has known anti-inflammatory function (*Broedl et al., 2007*) (p<0.05, Bonferroni-corrected one-sample *t*-tests; *Figure 1B*). Yet, other biomarkers did not associate antibiotic treatment with an immune response. *Nfkb1* expression, which is associated with inflammation suppression (*Cartwright et al., 2016*), increased after antibiotic treatment. Antibiotic treatment was also followed by a small, but significant, decrease in lipocalin-2 levels, which is a protein biomarker of inflammation (*Chassaing et al., 2012*) (p<0.001, linear mixed effects model likelihood tests; *Figure 1C*). Taken together, our biomarker assays provided equivocal evidence for intestinal inflammation in antibiotic-treated mice.

## Bacterial responses to antibiotics associated with redox potential shift

To test if inhibition of bacterial populations by antibiotics could directly contribute to redox potential dynamics, we used an ex vivo human gut system based on a continuous-flow bioreactor (*McDonald et al., 2013*), which allowed the propagation of a stable microbial community representative of the human gut microbiota with all major phyla represented (*Figure 2—figure supplement 1*). Treating this system with the same antibiotic cocktail as used in the mouse study led to an increase in redox potential relative to an untreated control (p=0.005, linear mixed effects model likelihood tests; *Figure 2A*). Redox potential increased by 59 ± 47 mV within fifteen hours of the first antibiotic dose, mirroring the rapidity with which redox shifts occurred in vivo (*Figure 1A*). Redox potential in our ex vivo model increased again by another 141 ± 37 mV after a second antibiotic dose (*Figure 2—figure supplement 1*). Thus, in the absence of direct interactions between

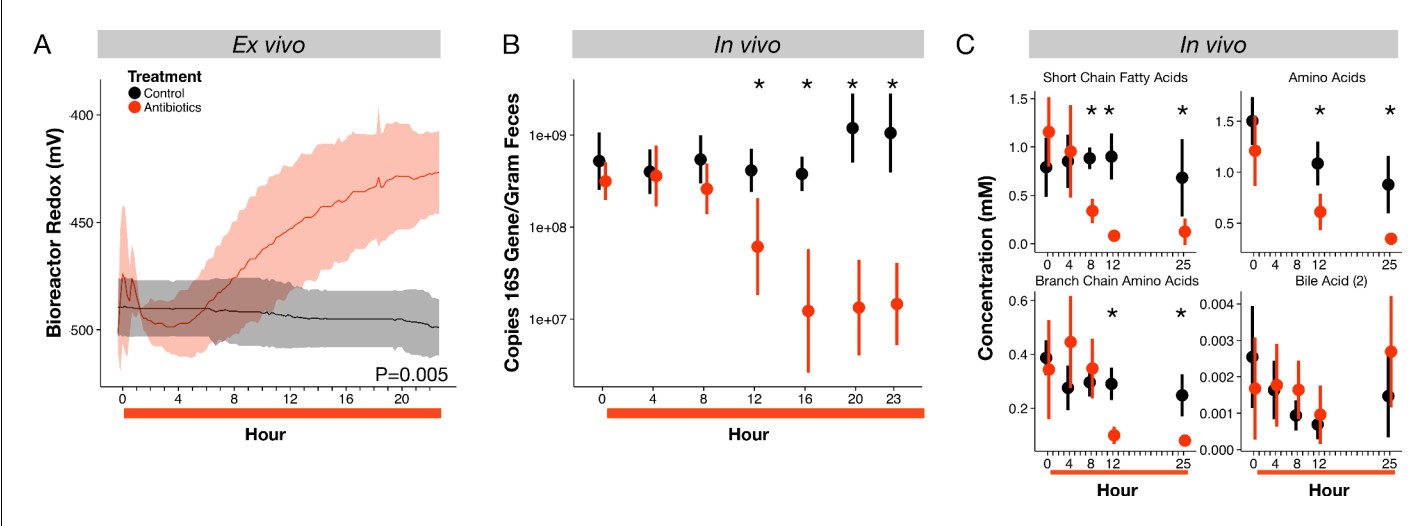

**Figure 2.** Microbial response to antibiotics ex vivo and in vivo. (**A**) Redox potential from control (black) and antibiotic treated (red) bioreactor vessels culturing human gut microbial communities (n = 3 per treatment) differed (p=0.005, linear mixed effects model likelihood tests). (**B**) Microbial load, measured as 16S rRNA gene copy number in antibiotic-treated and control mice decreased during the first 24 hr of treatment for antibiotic treated mice (n = 9–10 per treatment; p<0.001, Bonferroni-corrected Mann-Whitney U tests). (**C**) Metabolites measured with NMR spectrometry of feces (n = 9–10 per treatment; see *Figure 2—figure supplement 3* and *Supplementary file 1*) decreased in antibiotic treated mice during the first 24 hr (p<0.001, Bonferroni-corrected Mann-Whitney U tests). Data are shown as means ± SD. Post-hoc test results for individual time points (* indicates p<0.05 Bonferroni-corrected Mann-Whitney U test) are included for visualization purposes. Antibiotic treatment began after Day 0 measurement; red bars indicate treatment duration.

DOI: https://doi.org/10.7554/eLife.35987.006

The following figure supplements are available for figure 2:

**Figure supplement 1.** Ex vivo bioreactor experiments demonstrate microbial change alone can alter redox levels.

DOI: https://doi.org/10.7554/eLife.35987.007

**Figure supplement 2.** Overall antibiotic effects on bacterial load.

DOI: https://doi.org/10.7554/eLife.35987.008

**Figure supplement 3.** Overall antibiotic effects on metabolites.

DOI: https://doi.org/10.7554/eLife.35987.009

antibiotics and a host, antibiotic treatment produced shifts in environmental redox potential experienced by a gut microbial community.

## Fecal redox potential under antibiotics is associated with bacterial metabolism and respiration

To understand how antibiotic effects on bacterial populations could lead to shifts in gut redox potential, we investigated the dynamics and metabolism of microbiota across treatment in mice. We observed depressed levels of bacterial load and metabolic activity that occurred within hours of initial redox potential shifts. Fecal bacterial concentrations decreased significantly within twelve hours of antibiotic treatment (p=0.01, Bonferroni-corrected Mann-Whitney U test; *Figure 2B*), and remained significantly below controls throughout the five days of treatment (p<0.001, Bonferroni-corrected Mann-Whitney U tests; *Figure 2—figure supplement 2*). Using NMR-based metabolomics, we found that the short-chain fatty acids propionate and acetate, end products of key microbial metabolic pathways, decreased eight hours after antibiotic treatment (p<0.05, Bonferroni-corrected Mann-Whitney U tests; *Figure 2—figure supplement 3*, *Supplementary file 1*). Twenty-one of twenty-eight metabolites measured, including other short-chain fatty acids, amino acids, branched-chain amino acids, and a group of bile acids, later decreased significantly under antibiotic treatment (p<0.05, Bonferroni-corrected Mann-Whitney U tests; *Figure 2C*, *Figure 2—figure supplement 3*, *Supplementary file 1*). The dynamics of twelve of those metabolites was significantly associated with treatment overall (p<0.05, linear mixed effects model likelihood tests;

*Supplementary file 1*). Thus, the timing and persistence of decreased gut load and microbial activity coincided with increases in fecal redox potential.

Next, we measured the concentration of three major electron acceptors used during microbial respiration to assess whether changes in these pathways contributed to the redox potential increase. First, measuring luminal oxygen with a novel in vivo sensor system—a hydrogel sensor with covalently attached oxygen sensitive Pd-porphyrin derivative was inserted rectally then read optically through the skin—we observed a significant increase in luminal oxygen levels the day after antibiotic treatment ($3.7 \pm 3.6$ Torr at baseline to $26.5 \pm 26.1$ Torr; p=0.02, Bonferroni-corrected Mann-Whitney U test; *Figure 3A*). Previous work (*Kelly et al., 2015*; *Rivera-Chávez et al., 2016*) has shown elevated tissue oxygenation under antibiotics, which could result in higher diffusion and therefore higher luminal oxygen. In addition, reduced aerobic respiration due to antibiotic mortality or stress could also increase gut oxygen levels. However, gut oxygen levels in antibiotic-treated mice returned to control levels by forty-eight hours after the first dose (p>0.05, Bonferroni-corrected Mann-Whitney U test), and antibiotics were not associated with an overall effect on gut oxygen levels (p>0.05 linear mixed effects model likelihood tests). Moreover, among treated mice, gut oxygen levels were not correlated with redox potential (p>0.05, repeated measures correlation; *Figure 3—figure supplement 1*). These data together suggest that increased luminal oxygenation during antibiotic treatment may contribute to early shifts in redox potential, but were not responsible for persistent redox shifts.

By contrast, we observed sustained and significant increases in nitrate levels during antibiotic treatment. Nitrate is one of the most widely used electron acceptors in the gut (*Ravcheev and Thiele, 2014*; *Fischer and Lindenmayer, 2007*) and its reduction to nitrogen gas (at pH 7) has a potential of +0.75 V making it one of the most potent electron acceptors after oxygen. The increase in *Nos2* expression (*Figure 1B*) led us to hypothesize that nitrate would increase as it is a gene which encodes inducible nitric oxide synthase and whose expression has been found to confer a growth

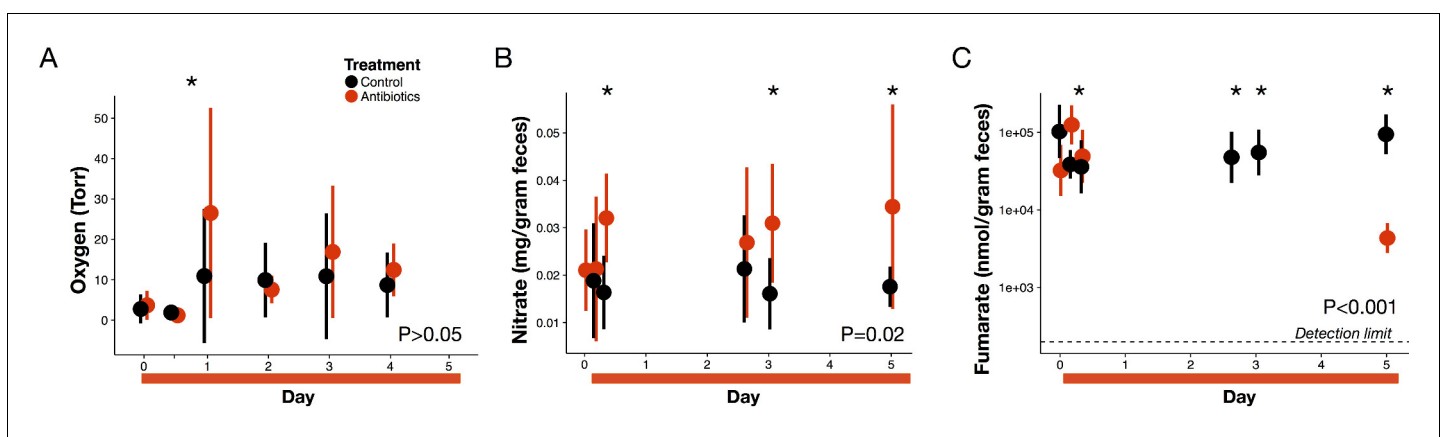

**Figure 3.** Electron acceptor levels during antibiotic treatment. (**A**) Large intestine luminal oxygen concentration measured in vivo for treated (red) and control (black) mice (n = 9–10 per treatment) did not differ overall during treatment (p>0.05, linear mixed effects model likelihood tests). Replicate data are presented in *Figure 3—figure supplement 2*. (**B**) Fecal nitrate concentration (n = 2–9 per group per time point) did differ between treated and control mice during treatment (p=0.02, linear mixed effects model likelihood tests). (**C**) Fecal fumarate concentration (n = 2–9 per group per time point) also differed (p<0.001, linear mixed effects model likelihood tests). Electron acceptor levels below the detection level are not plotted here. For fumarate measurements where all treated animals were below the detection limit, values were set to 200 nmol for statistical tests. Data are shown as means ± SD. Post-hoc test results for individual time points (* indicates p<0.05 Bonferroni-corrected Mann-Whitney U test) are included for visualization purposes. Antibiotic treatment began after Day 0 measurement; red bars indicate treatment duration.

DOI: https://doi.org/10.7554/eLife.35987.010

The following figure supplements are available for figure 3:

**Figure supplement 1.** Redox level is not driven by oxygen concentration.
DOI: https://doi.org/10.7554/eLife.35987.011
**Figure supplement 2.** Overall antibiotic effects on oxygen.
DOI: https://doi.org/10.7554/eLife.35987.012
**Figure supplement 3.** Oxygen data during treatment from replicate experimental run.
DOI: https://doi.org/10.7554/eLife.35987.013

advantage to nitrate respiration competent *E. coli* strain (*Winter et al., 2013*). Lower levels of microbial respiration could also lead to an accumulation in nitrate separate from changes in host expression. We measured fecal nitrate levels with the chromotropic acid method (*West and Ramachandran, 1966*) for a subset of time points when redox varied, and found a significant increase in nitrate (p=0.03, Mann-Whitney U test; *Figure 3B*) beginning less than 24 hr after the first dose of antibiotics. This increase persisted throughout treatment, and overall antibiotic treatment produced a significant increase in nitrate (p=0.02, linear mixed effects model likelihood tests).

Finally, we used a colorimetric enzyme assay to measure fecal fumarate levels for a subset of time points during antibiotic treatment. Fumarate reductases have been found in over one third of gut microbe genomes, and fumarate is the most common terminal electron acceptor for bacterial anaerobic respiration (*Kröger et al., 1992*). Unlike nitrate and oxygen, though, much of the fumarate in the gut is likely produced by the microbiota itself (*Fischbach and Sonnenburg, 2011*; *El Aidy et al., 2013*). Overall, there was a significant effect of treatment on fumarate levels (p<0.001, linear mixed effects model likelihood tests; *Figure 3C*). We observed a spike in fumarate levels 4 hr after the first antibiotics dose (p=0.004, Bonferroni-corrected Mann-Whitney U test); however, this increase had dissipated by 8 hr after the first dose and later during treatment, we observed a significant decrease (to below the detection limit) in fumarate levels. Thus, not all electron acceptors increased in abundance during antibiotic treatment.

## Community composition and gut environment recover following antibiotic treatment

Ecological succession, the 'somewhat orderly and predictable' (*Fierer et al., 2010*) dynamics of a community after a perturbation, is known to follow antibiotic treatment (*Antonopoulos et al., 2009*; *David et al., 2015*). What is less well-known is how abiotic conditions change during post-antibiotic succession and, by extension, how biotic and abiotic dynamics interface during that period. Having shown that both the community and gut environment are not resistant to antibiotic disturbance, we next sought to determine how resilient they were, *i.e.* how quickly they recovered (*Shade et al., 2012*; *Allison and Martiny, 2008*). In this study, we indeed observed consistent biotic changes over time in mice that received antibiotics (*Figure 4—figure supplement 1*). Gut bacterial load recovered rapidly after treatment across all treated mice, exhibiting no difference relative to control mice by two days (p>0.05, Bonferroni-corrected Mann-Whitney U tests; *Figure 2—figure supplement 2*). A reproducible recovery of beta-diversity was observed after one week (p>0.05, Bonferroni-corrected Mann-Whitney U test; *Figure 4—figure supplement 2*).

To characterize the recovery of specific microbial taxa, we clustered fecal bacterial genera into groups according to their dynamical patterns across antibiotic treatment (*Figure 4A,B*). We focused on the five clusters with an average abundance of at least 1% across our dataset (*Supplementary file 2*). Amongst these five groups, three differed significantly during treatment (p<0.05, linear mixed effects model likelihood tests; *Figure 4A,B*). Notably, the two clusters elevated during treatment included many facultative anaerobic taxa (*Supplementary file 2*). One of these, the cluster primarily composed of members of the Enterobacteriaceae, remained elevated into the beginning of the recovery period as well (p<0.05, Bonferroni-corrected Mann-Whitney U tests). This enrichment is in line with previous findings that Enterobacteriaceae often bloom following antibiotic treatment (*Antonopoulos et al., 2009*; *David et al., 2015*; *Young and Schmidt, 2004*; *Theriot et al., 2014*; *Peterfreund et al., 2012*; *Dethlefsen et al., 2008*; *Jakobsson et al., 2010*; *Looft and Allen, 2012*). That cluster, as well as one composed primarily of *Akkermansia* and one of a consortium of typical commensal taxa, exhibited significant residual effects of treatment during the recovery period (p<0.05, linear mixed effects model likelihood tests; *Figure 4A,B*). However, by the end of the one week recovery period, the abundance of all clusters was indistinguishable between treated and control animals (p>0.05, Bonferroni-corrected Mann-Whitney U tests).

Abiotic conditions in the mouse gut were also resilient and recovery was predictable. Redox potential returned to control levels within a week after treatment ended (p>0.05, Bonferroni-corrected Mann-Whitney U tests; *Figure 4C*). Notably, though, we observed the day after antibiotic therapy ceased, fecal redox potentials in treated mice were significantly decreased relative to control mice (-138.3 ± 149.8 mV vs 54.6 ± 211 mV; p=0.006, Bonferroni-corrected Mann-Whitney U test), and it was not until later in recovery that redox potential returned to control levels across most mice. Oxygen and inflammation biomarker levels remained low throughout the recovery period

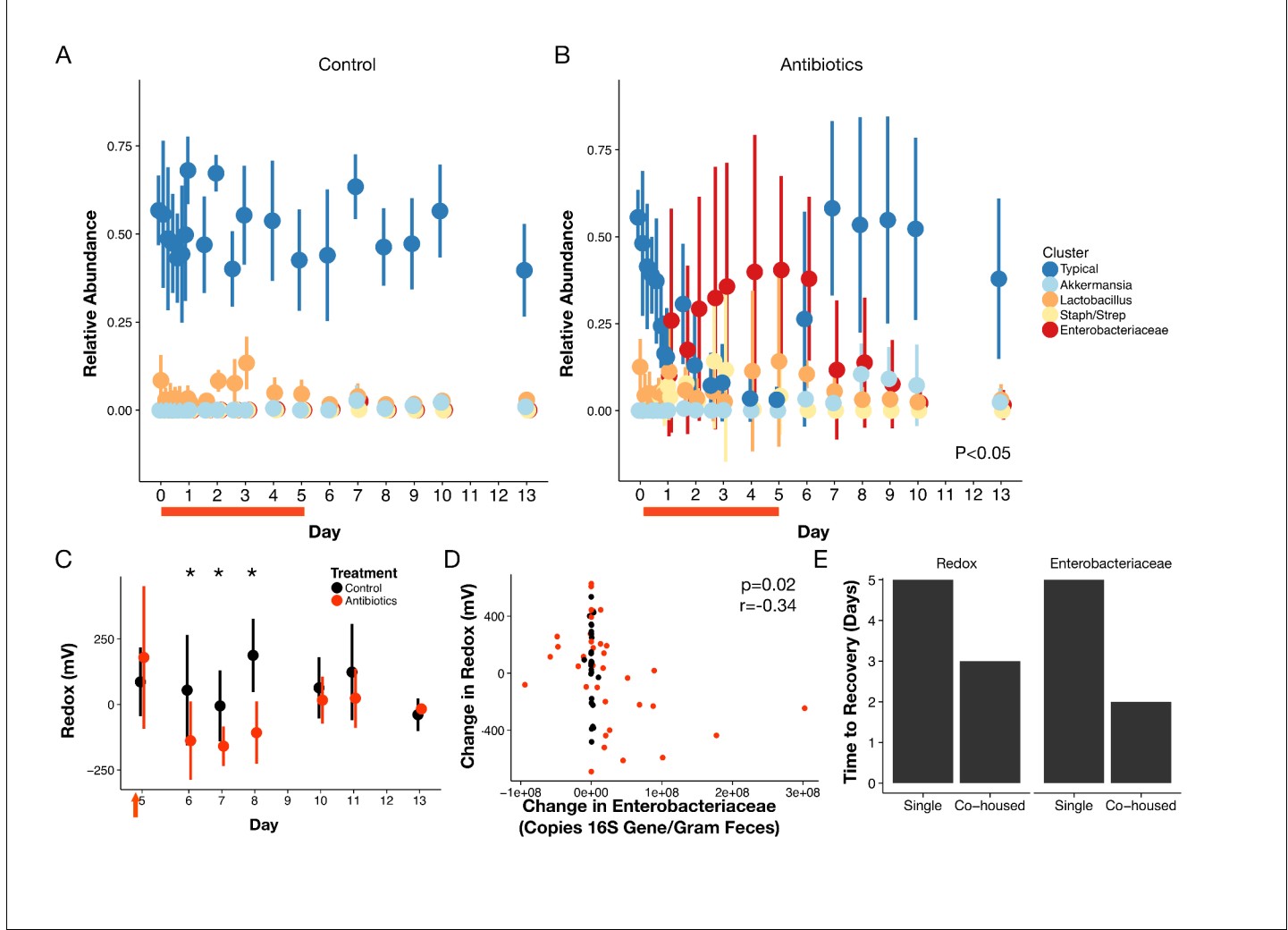

**Figure 4.** Gut community is resilient to antibiotic disturbance. (**A–B**) Compositional dynamics of most abundant clusters during and after treatment for control (**A**) and treated (**B**) mice (n = 9–10 per treatment). Three groups (Typical, Staph/Strep, Enterobacteriaceae) differed significantly between groups (p<0.05, linear mixed effects model likelihood tests) during treatment and two (Enterobacteriaceae, Akkermansia) during recovery. (**C**) Redox potential measured in freshly voided feces of singly housed mice during the recovery period (n = 5 per treatment) was lower than treated mice initially (p<0.05, Bonferroni-corrected Mann-Whitney U tests). (**D**) Repeated measures correlation was significant (p=0.02) between change in Enterobacteriaceae abundance and change in redox potential during the first three days of post-antibiotic recovery (n = 64). (**E**) Days until there is no significant difference (p>0.05 Bonferroni-corrected Mann Whitney U test) between groups of control and treated animals for redox and Enterobacteriaceae abundance, under either singly- housed or co-housed settings. Red bars indicate treatment duration (**A–B**). Red arrows indicate the last antibiotic dose; day six measurements are more than 24 hr after the last dose (**C**). Data are shown as means ± SD. Post-hoc test results for individual time points (* indicates p<0.05 Bonferroni-corrected Mann-Whitney U test) are included for visualization purposes.

DOI: https://doi.org/10.7554/eLife.35987.014

The following figure supplements are available for figure 4:

**Figure supplement 1.** Antibiotic driven community composition changes.
DOI: https://doi.org/10.7554/eLife.35987.015

**Figure supplement 2.** Overall antibiotic effects on beta-diversity.
DOI: https://doi.org/10.7554/eLife.35987.016

**Figure supplement 3.** Abundance of the Enterobacteriaceae is associated with abiotic conditions and changes in redox state.
DOI: https://doi.org/10.7554/eLife.35987.017

**Figure supplement 4.** Post-antibiotic succession can occur within one week but is mediated by cohousing status of mice.
DOI: https://doi.org/10.7554/eLife.35987.018

**Figure supplement 5.** Mouse cohorts differ in original community state but not response trends.
DOI: https://doi.org/10.7554/eLife.35987.019

(p>0.05, Bonferroni-corrected Mann-Whitney U tests; *Figure 3—figure supplement 2*, *Figure 1—figure supplement 1*, respectively) consistent with their minimal changes during treatment. Furthermore, many microbial metabolic products measured by NMR had recovered two days after the last doses, and all had recovered by the end of the recovery week (*Figure 2—figure supplement 3*; *Supplementary file 1*). In concert, a largely reproducible biotic and abiotic recovery took place across antibiotic treated mice in the days following antibiotic treatment indicating the overall resilience of this ecosystem to antibiotic disturbance.

## Post-antibiotic succession resulted from interplay of biotic and abiotic recovery

In investigating potential interactions between abiotic and biotic processes in antibiotic recovery, we hypothesized that the Enterobacteriaceae could thrive under high redox potential conditions and directly impact gut redox state. Members of this bacterial family were enriched at the end of treatment in our clustering analysis (*Figure 4B*). Enterobacteriaceae can also employ many terminal electron acceptors to perform aerobic and anaerobic respiration (*Ravcheev and Thiele, 2014*) and thus could likely thrive under high redox potential conditions and directly impact gut redox state. We found that the relative abundance of the Enterobacteriaceae was marginally correlated with redox potential (p=0.1, r=,0.18, repeated measures correlation; *Figure 4—figure supplement 3*) throughout treatment and recovery. Moreover, the change in redox potential during the earlier recovery period was negatively correlated with changes in Enterobacteriaceae absolute abundance (calculated as total 16S rRNA gene copy number multiplied by relative abundance) for both treated animals (p=0.03, r = −0.42, repeated measures correlation) and for all animals (p=0.02, r = −0.34, repeated measures correlation; *Figure 4D*). Together, these observations suggest that increased respiration by the Enterobacteriaceae after antibiotic treatment may lead redox potentials to decrease below conventional levels. More broadly, our observations support the hypothesis that Enterobacteriaceae dynamics contribute to natural redox potential variation in the gut.

We next investigated if abiotic recovery led to compositional recovery or if dispersal (*i.e.*, migration of microbes) from unimpacted microbial populations was necessary for microbiota to exhibit resilience and return to a pre-treatment state. Previous research has shown that singly housed mice treated with antibiotics will exhibit altered community composition for weeks after treatment (*Antonopoulos et al., 2009*). We therefore co-housed half of the treated mice with control animals during the recovery period and kept half in single housing. Because mice practice coprophagy (*i.e.*, ingestion of feces), we would expect that co-housing would re-introduce normal commensal microbes to compete with the Enterobacteriaceae and other early successional species thereby reducing their abundance. At the end of the one-week treatment period, we indeed observed differential success in community recovery between the housing groups. Co-housed mice communities had returned to control composition while singly housed mice had not. For co-housed mice, by the end of the recovery period, the gut microbiota of treated animals was equally dissimilar from baseline as it was for control mice indicating that the community had recovered (p>0.05, Bonferroni-corrected Mann-Whitney U test; *Figure 4—figure supplement 4*). Both fecal redox potential and Enterobacteriaceae levels also returned to conventional levels more quickly in co-housed mice than singly housed mice (*Figure 4E*). By contrast, singly-housed treated mice continued to be more dissimilar at the end of recovery than singly-housed control mice (p<0.001, Bonferroni-corrected Mann-Whitney U test; *Figure 4—figure supplement 4*). Such drift is qualitatively consistent with arguments that stochastic variation can influence how communities respond to perturbation, as seen in soil microbial communities (*Zhang et al., 2016*). More broadly, the differences between singly- and co-housed mice here highlight that environmental recovery is not sufficient to produce biotic recovery—dispersal is also necessary for a resilient community.

## Discussion

Here, we found that a key feature of the microbial environment, redox potential, can be rapidly and dramatically altered in the gut under antibiotic treatment. This shift was accompanied by changes in electron acceptor availability. The direction and timing of the redox potential shift was associated with microbial dynamics and could be reproduced with an ex vivo model of the human gut absent host effects. Moreover, we found evidence that redox potential kinetics were driven by shifts in

microbial load and metabolism, specifically among facultative anaerobes from the family Enterobacteriaceae. Although there was evidence of increases in certain host pathways of electron acceptor production, we did not find persistent and significant changes in host inflammation. Therefore, we propose microbial respiration of host sourced electron acceptors acts as a major determinant of redox potential in the gut, in addition to the host production itself. Redox potential consequently shaped successional dynamics following the cessation of treatment, but return to control redox conditions was insufficient to produce community recovery. Dispersal was required for community structure to exhibit full resilience to antibiotic disturbance.

Our findings imply that electron acceptors are a critical resource normally competed over by the resident bacteria and, as such, are key factors structuring gut microbial ecology. The ability to respire a broad range of electron acceptors is already known to be crucial for pathogens to colonize the intestine (*Winter et al., 2010*; *2013*; *Faber et al., 2016*). Yet, ecological research to date on commensal microbiota has primarily focused on substrate availability (*Donaldson et al., 2016* and references therein, *Pereira and Berry, 2017*) or, to a lesser extent, oxygen (*Espey, 2013*). Our results here support the hypothesis that the availability of other electron acceptors is an important ecological factor driving the structure of resident bacterial populations as well (*Jones et al., 2007*; *2011*). If true, this hypothesis would imply that pathogen colonization resistance by commensal microbes could be mediated by bacterial competition involving respiratory pathways.

Future work will be needed to comprehensively chart the dynamics and chemical landscape across which bacteria in the gut compete for electron acceptors. While any redox active molecule could contribute to redox potential changes (hence testing all possible chemical drivers of redox is infeasible), we focused here on three relevant electron acceptor species (oxygen, nitrate, and fumarate) that derive from independent pathways and may be representative of overall dynamics. Our data lead us to conjecture that oxygen availability transiently increased due to the decreased respiration capacity of an antibiotic-constrained bacterial community. But, as oxygen grew increasingly available, the remaining facultative anaerobes switched to aerobic respiration leaving other host-sourced electron acceptors, such as nitrate, to accumulate. Accrued electron acceptors, especially nitrate, may also have been contributed by changes in host physiology as evidenced by an upregulation of the *Nos2* gene, which is linked to reactive nitrogen species levels (*Dedon and Tannenbaum, 2004*; *Winterbourn, 2008*). Still, not all electron acceptors accumulated during antibiotic treatment. Notably, our data suggest that ones produced by the microbiota rather than the host, like fumarate, were less available under antibiotic pressure.

Our data also support a model where consumption by specific bacterial taxa shape redox homeostasis in the gut. We observe that facultative anaerobes like the Enterobacteriaceae behave as pioneer taxa whose dynamics inversely track redox dynamics after antibiotic treatment, providing new understanding of the deterministic mechanism driving the consistent bloom of these bacteria following antibiotic treatment (*Antonopoulos et al., 2009*; *David et al., 2015*; *Young and Schmidt, 2004*; *Theriot et al., 2014*; *Peterfreund et al., 2012*; *Dethlefsen et al., 2008*; *Jakobsson et al., 2010*; *Looft and Allen, 2012*). Enterobacteriaceae are especially suited to exploiting electron acceptor availability because their diverse repertoire of respiration pathways (*Ravcheev and Thiele, 2014*; *Jones et al., 2007*; *2011*) allow for fast growth rates under a high redox regime. Enterobacteriaceae and respiration pathway genes more generally are also more prevalent in inflamed mouse guts (*Hughes et al., 2017*), another environment where heightened redox potential is expected. Under experimental inflammation, though, Enterobacteriaceae expansion can be prevented by treatment with tungstate, a respiration inhibitor (*Zhu et al., 2018*). This result corroborates our model that members of the Enterobacteriaceae use respiration to bloom under disturbed conditions, and it highlights the potential for manipulating electron acceptor availability or usability to engineer the microbiota.

An increase in resource availability is common following ecological disturbance and can promote the growth of pioneer taxa, which are best suited to quickly respond to a resource spike but which draw down the resource thereby bringing about their own displacement (*Connell and Slatyer, 1977*; *Peet and Christensen, 1980*; *Tilman, 1985*). Here, as in many cases of ecological succession, the Enterobacteriaceae were eventually replaced by more conventional community members that typically grow better under more limiting conditions; in fact, the relationship between change in Enterobacteriaceae abundance and change in redox state is consistent with these taxa facilitating the success of secondary colonizers. However, why redox recovery was ultimately hysteretic (*i.e.*, the gut temporarily became a reducing environment following the end of treatment) remains unclear. It

has previously been shown that members of the Enterobacteriaceae will produce reducing agents, such as $H_2S$, in response to antibiotic treatment (*Shatalin et al., 2011*), but this theory requires further investigation in a community and host-associated context.

Beyond identifying a role for microbiota in shaping gut redox potentials, our results more broadly illustrate how microbial factors can be important in addition to host factors for determining abiotic conditions in the gut through the balancing of production and consumption of resources. While the host is ultimately the source of all substrates in the gut, microbial action determines the realized environment which shapes community composition and function. Such microbial ecosystem engineering (*Jones et al., 1994*; *Wright and Jones, 2006*) is consistent with our understanding of how microbes can shape their environments in free-living systems (*Gerbersdorf et al., 2009*; *Goddard, 2008*). The altered resource availabilities that result from microbial action then contribute to structuring the microbial community itself. Such feedbacks may contribute to a number of phenomena in host-associated systems. Common constituents of the gut (*i.e.*, the phyla Firmicutes and Bacteroidetes) could perpetuate similar hospitable environments for themselves regardless of host biology leading to the broad, phylum-level consistency of composition seen between humans regardless of diet, geography, or genetics (*David et al., 2014*; *Claesson et al., 2012*; *Goodrich et al., 2014*; *De Filippo et al., 2010*; *Yatsunenko et al., 2012*; *Arumugam et al., 2011*). Variations in how the gut environment is dictated by microbes may also help explain inter-individual differences in response to interventions in the absence of consistent host effects (e.g., *Venkataraman et al., 2016*). Microbial metabolic signaling additionally can alter host production of substrates including redox active molecules. Most notably reduced butyrate levels, a change observed here, have been demonstrated to increase gut nitrate levels (*Byndloss et al., 2017*) and host epithelial oxygenation (*Rivera-Chávez et al., 2016*).

## Conclusion

Together, our findings suggest new ecological models for how antibiotics reshape the gut microbiota and for how redox shifts could be associated with enteric disease. Antibiotics are triumphs of modern medicine that have dramatically reduced infectious disease mortality (*Armstrong et al., 1999*). But, we are increasingly learning that antibiotics also meaningfully reshape the resident gut microbiota, leaving an imprint that can last for months to years after treatment (*Dethlefsen and Relman, 2011*; *Jakobsson et al., 2010*) and predisposing hosts to obesity (*Cho et al., 2012*), food allergy (*Stefka et al., 2014*), auto-immune disease (*Russell et al., 2012*), and increased infection risk (*Stecher et al., 2007*; *Buffie et al., 2012*; *Wiström et al., 2001*). While these drugs reduce levels of susceptible organisms (*Keeney et al., 2014*), an additional ecological mechanism of action is decreasing microbial competition and allowing primary metabolites (*e.g.*, primary bile acids, sugars) (*Ng et al., 2013*; *Theriot et al., 2014*), as well as host-sourced electron acceptors like oxygen and nitrate to accumulate. This concept complements recent discoveries that electron acceptors facilitate antibiotic-associated enteric pathogen colonization (*Rivera-Chávez et al., 2016*; *Winter et al., 2013*; *2010*). Such increases in electron acceptor availability likely are not unique to antibiotic treatment and could generalize to various enteric disturbances. Indeed, germ-free animals (*Phillips et al., 1958*; *Celesk et al., 1976*) and humans suffering from inflammatory diseases (*Circu and Aw, 2011*) and malnutrition (*Million et al., 2016*) exhibit increased gut redox potential. Thus, novel treatments for microbial disorders or preventing antibiotic-associated infections may include chemical alterations of redox potential or introduction of competitors for excess electron acceptors. More broadly, we propose adding redox potential to the list of abiotic conditions frequently assayed and manipulated to improve host well-being.

## Materials and methods

### Animal experiments

#### Mice

All animal experiments were conducted in accordance with National Institute of Health Guide for the Care and Use of Laboratory Animals using protocols approved by the Duke University Institutional Animal Care and Use Committee. Male C57BL/6 mice (Charles River Laboratories, Wilmington MA) 8–10 weeks of age with a native microbiota were used for all experiments. Mice were kept in a conventional laboratory animal facility at Duke University. Baseline measurements and fecal samples

were collected at least twenty-four hours before the first dose and then mice were placed in individual housing with supplementary enrichment.

## Antibiotic treatment

Mice were orally gavaged with either 0.25 ml autoclaved deionized water (control, N = 10) or 0.25 ml of an antibiotic cocktail (treated, N = 10) daily for five days (*Figure 5A*). The mice were randomly assigned a group and researchers collecting data were blinded to the groupings until after the final dose was administered. The sample size was chosen following a power analysis to allow for β less than 0.1. The antibiotic cocktail consisted of ampicillin (Gold Biotechnology, St. Louis MO) 1 mg/ml, vancomycin (Alfa Aesar, Ward Hill MA) 5 mg/ml, neomycin (EMD Millipore, Burlington MA) 10 mg/ml, and metronidazole (Alfa Aesar) 10 mg/ml (after *Reikvam et al., 2011*). Fresh antibiotic cocktails were prepared every day. Throughout the experiment freshly voided fecal samples were collected and stored at −80°C for subsequent analysis.

## Recovery cohousing

One day after the final gavage, mice were randomly assigned to cohousing groups (*Figure 5B*). Cohousing control cages (N = 8, four each from treated and control) contained single mice that were kept in individual housing throughout the recovery period. Cohousing treatment cages (N = 5) contained a control and an antibiotic-treated mouse from the same litter that were placed together in the control mouse cage and kept together throughout the recovery period. The sample size was determined by the number of mice available at the end of the treatment period. Antibiotic-treated mice in cohoused cages were marked by partial shaving to allow for continued sampling. Throughout recovery, freshly voided fecal samples were collected and stored at −80°C for subsequent analysis. While the antibiotic treatment experiment was performed multiple times, the cohousing during recovery experiment was only performed with a single replicate cohort.

## Redox potential measurements

We measured redox potential in fresh fecal pellets using a redox electrode with a tip diameter of 500 μm in tandem with an Ag/AgCl reference electrode (Unisense, Aarhus Denmark). The values are

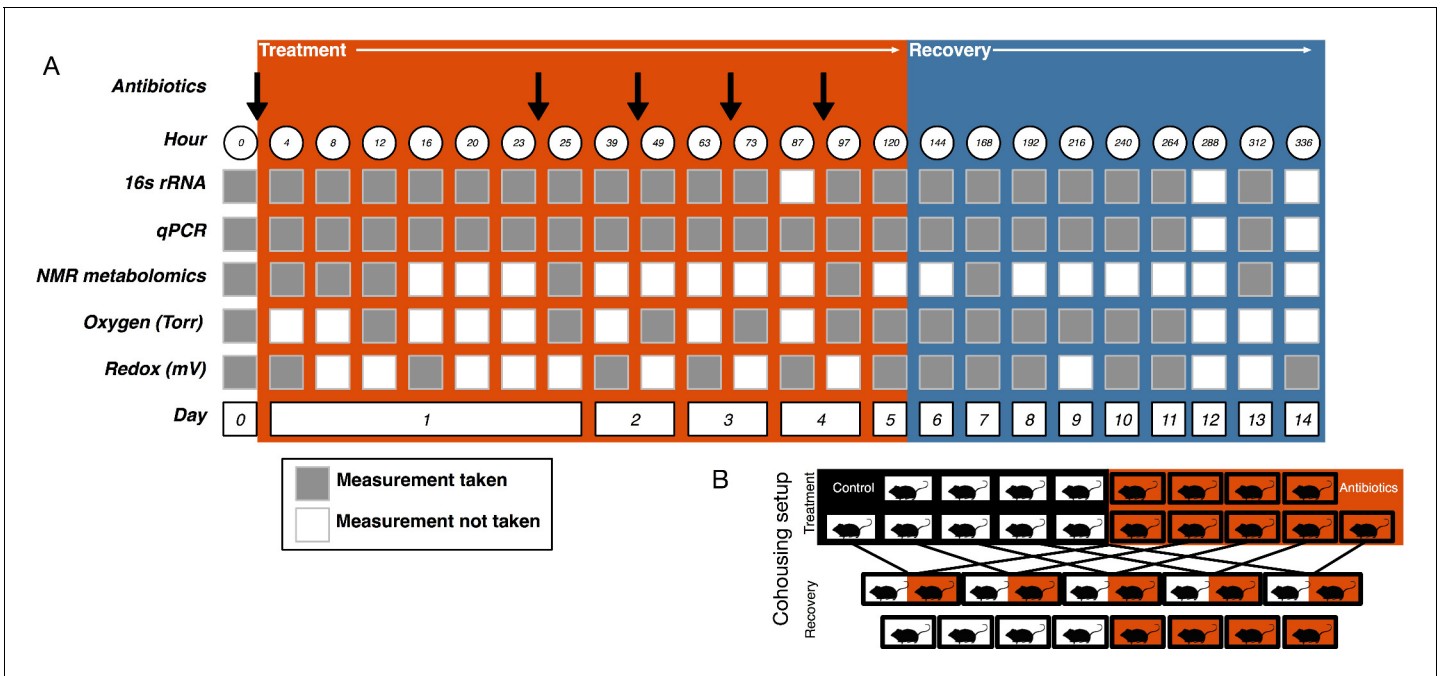

**Figure 5.** Experimental setup. (**A**) Experimental sampling regime. (**B**) Cohousing treatment setup.
DOI: https://doi.org/10.7554/eLife.35987.020

given relative to the standard hydrogen electrode (SHE) and were determined by measuring the off-set of the reference electrode in saturated quinhydrone buffer solutions (pH 4 and pH 7) with known redox potentials. Daily calibrations were done at room temperature; however, because of potential unmeasured variation in calibration procedures (*e.g.*, fluctuation in ambient temperature), we focused our analyses on differences between control and treated animals at any given time point. The electrode has a detection limit of 0.10 mV (*Pang and Zhang, 1998*).

Fecal pellets were placed on an agar plate, resting against the reference electrode. The electrode was inserted into the pellet using a micromanipulator and data were collected continuously for three minutes. The data presented are the average across three minutes. Redox data was collected on two cohorts of mice with comparable results. Replicate data is shown in *Figure 1—figure supplement 2*.

## Oxygen measurements

We measured oxygen concentration in vivo using a hydrogel oxygen microsensor coupled with an optical reader. The microsensors are composed of a biocompatible hydrogel, poly (2-hydroxyethyl methacrylate) (pHEMA) and a near infrared (NIR) oxygen-sensitive palladium-benzoporphyrin mole-cule (Pd-MABP) (*Montero-Baker et al., 2015*; *Wisniewski et al., 2017*; *Chien et al., 2017*). The microsensor measures oxygen based on the principle of phosphorescence quenching of metallopor-phyrins, a well-established technique with excellent sensitivity and specificity to physiologic oxygen (*Wilson et al., 2006*; *Rumsey et al., 1988*; *Lo et al., 1996*; *Vinogradov et al., 2003*). The pHEMA hydrogel is biocompatible, has good oxygen permeability, excellent mechanical properties, and a long history of use in medical devices (*Montheard et al., 1992*). The Pd-MABP molecules are cova-lently attached to the pHEMA hydrogel, ensuring that the sensing chemistry is retained in the hydro-gel structure. The miniature sensors (0.75 x 0.75 x 2.5 mm) are soft and tissue-like to minimize stress at the material-tissue interface.

The non-invasive optical reader was manually positioned over the sensor and a LED pulsed illumi-nation light into the skin above the sensor. A photodetector collected emission light emanating from the sensor. The phosphorescent lifetime, a property of the oxygen sensitive dye, was measured, thereby providing a signal unaffected by optical path permutations such as sensor depth, blood flow fluctuations, oxy/deoxyhemoglobin ratio, melanin content and hydration (*Montheard et al., 1992*). The current oxygen sensing system measures lifetime to within 2 µs or better, which equates to ~1 µM $O_2$ at physiological oxygen conditions. The temperature was assumed to be at 37°C to convert the lifetime measurement to oxygen concentration.

The oxygen sensor was placed via rectal insertion in the distal large intestine (~1 cm from anus) of mice under isofluorane anesthesia. The phosphorescent signal was collected for two minutes using the optical reader and all values were averaged before conversion to oxygen concentration. Upon reawakening from anesthesia, the sensor was passed naturally from the mouse via peristalsis. Oxy-gen data were collected on two cohorts of mice with comparable results. Replicate data is shown in *Figure 3—figure supplement 3*.

## Nitrate measurements

We measured nitrate in frozen mouse fecal samples using the NitraVer X Nitrogen-Nitrate Reaction Set (Hach Company, Loveland CO) following the manufacturer's instructions. Data below the lower detection limit of 0.00036 mg were not included in plots or statistical tests. These measurements were collected for only a single cohort of mice due to availability of samples.

## Fumarate measurements

We measured fumarate in frozen mouse fecal samples using the Fumarate Colorimetric Assay Kit (Bio-vision, Milpitas CA) following the manufacturer's instructions. Data points below the lower detection limit of 200 nmol were not shown on plots, but were included in post-hoc statistical analyses as a pseudo-count of 200 nmol when all measures for a treatment group were below the detection limit. These measurements were collected for only a single cohort of mice due to availability of samples.

## DNA isolation from mouse fecal samples

Metagenomic DNA was isolated from frozen fecal samples using the MoBio (now Qiagen, Hilden Germany) PowerSoil DNA extraction kit following the manufacturer's instructions (*David et al., 2014*; *2015*).

## 16S rRNA gene sequencing and processing

We performed 16S rRNA gene amplicon sequencing using custom barcoded primers targeting the V4 region of the gene (*Caporaso et al., 2011*) and published protocols (*Caporaso et al., 2011*; *2012*; *Maurice et al., 2013*). Sequencing was conducted on an Illumina (San Diego, CA) MiSeq with paired end 250 bp reads using the V3 kit. All samples with fewer than 5000 reads were discarded. Sequencing data were processed using QIIME (version 1.8) (*Caporaso et al., 2010*) to produce an OTU table with 97% cluster similarity. Sequencing was performed on samples from two cohorts of mice with comparable results, but all results presented here are from a single cohort matching the abiotic data presented.

Because baseline microbiome is expected to differ between mouse litters over time as well as between vendor (*Rosshart et al., 2017*; *Rausch et al., 2016*; *Ivanov et al., 2009*; *Campbell et al., 2012*), we examined whether the effects observed were reproducible between the two cohorts sequenced. An overall analysis of beta-diversity showed that there were significant differences between cohorts in their microbiome at the OTU level throughout the experiment ($p < 0.01$, $R^2 = 0.17$, PERMANOVA; *Figure 4—figure supplement 5*). This cohort effect did not significantly interact with the treatment effect, however ($p = 0.3$). Because the Enterobacteriaceae bloom response to antibiotics has been found to be consistent between humans and mice, as well as between mouse vendors (*Antonopoulos et al., 2009*; *David et al., 2015*; *Young and Schmidt, 2004*; *Theriot et al., 2014*; *Peterfreund et al., 2012*; *Dethlefsen et al., 2008*; *Jakobsson et al., 2010*; *Looft and Allen, 2012*), we believe that the overall trends observed here are indicative of what may be occurring in other conventional gut microbiota contexts.

## Clustering

Bacterial genera were clustered using a previously described bioinformatics pipeline for identifying taxa with similar dynamics (*David et al., 2014*; *2015*). Rare genera (defined as those observed in five or fewer samples) were also excluded from clustering. Rare genera comprised 194 of 306 total observed genera, but accounted for only 0.019% of reads. All remaining genera ($n = 112$) were ultimately assigned to a cluster. A clustering threshold of 0.9 was manually chosen to balance model simplicity (*i.e.*, building clusters with an interpretable number of genera) and fidelity (*i.e.*, capturing the unique dynamics of distinct genera). Only clusters with an average abundance of >1% (five clusters altogether) were included in statistical analyses.

## qPCR: Primers and conditions

To estimate total bacterial abundance, PCR was performed on DNA extracted from feces using the following primers: forward, 5'-ACTCCTACGGGAGGCAGCAGT-3', reverse, 5'-GTATTACCGCGGC TGCTGGCAC-3' (*Bergström et al., 2012*). qPCR assays were run using SYBR FAST qPCR Master Mix (KAPA, Wilmington MA) on a 7900HT Fast Real-Time PCR System (Applied Biosystems, Foster City CA). Ct values were standardized against a dilution curve of known concentration and then adjusted for the weight of fecal matter extracted. Load measurements were performed on samples from two cohorts of mice with comparable results, but all data included here is from a single cohort.

## Inflammation assays

Fecal lipocalin 2 (Lcn-2) was used as a noninvasive biomarker for host intestinal inflammation. Lcn-2 was quantified by ELISA following Chassaing et al. (*Chassaing et al., 2012*). Assays were conducted with mouse lipocalin-2/ngal duo set ELISA kit from R and D Systems (Minneapolis, MN) using a Bio-Tek ELx405rs plate washer and a BioTek Synergy HT plate reader (Bio-Rad, Hercules CA) at the UNC Center for Gastrointestinal Biology and Disease Advanced Analytics Core. These measurements were collected for only a single cohort of mice due to availability of samples.

## RNA isolation and RT-PCR

Total RNA was isolated from fecal pellets stored in RNALater (Thermo Fisher, Waltham, MA) using the MoBio PowerMicrobiome RNA Isolation kit with an added phenol chloroform extraction step. $\leq$15 ng RNA was reverse transcribed using cDNA Prep Reverse Transcription Master Mix (Fluidigm, South San Francisco, CA) following manufacturer's instructions. Target transcripts were pre-amplified for 18 cycles and then diluted 10x. RT-PCR was performed using a BioMark (Fluidigm) on a 48 $\times$ 48 chip with Taqman Fast Advanced Master Mix (Thermo Fisher). Three ERCC RNA Spike-in Mix (Thermo Fisher) positive controls and a nontarget negative control of nuclease-free water were also run on the chip. Expression levels were quantified for thirty genes (*Supplementary file 3*). Of these, six genes could be quantified for at least 20 time points, and so only these genes were included in statistical analyses. Expression levels were normalized to mouse *Actb* ($\Delta$Ct) expression for each time point for each mouse and then compared to the average of control mice for that time point ($\Delta\Delta$Ct). Fold change ($2\Delta\Delta$Ct) is represented. These measurements were collected for only a single cohort of mice due to availability of samples.

## NMR metabolomics

Samples were prepared by mixing fecal pellets with 200 µl phosphate buffered saline (PBS) prepared in 95% $D_2O$, pH 7.5 and homogenizing for 5 min at 30 Hz. Cell debris was removed by centrifuging the samples for 10 min at 14K x g at 4°C. The supernatant was filtered through a 0.22 µm centrifugal filter. To maximize the metabolite extraction, each pellet from the first centrifugation was mixed with 200 µl PBS in 95% $D_2O$, pH 7.5 and homogenized for 5 min at 30 Hz. The solutions were filtered through a 0.22 µm centrifugal filter as before. The supernatants were combined and 300 µl was taken and mixed with 300 µl of 99.9% $D_2O$ containing 500 µM 3-(trimethylsilyl)−1-propanesulfonic acid-$d_6$, sodium salt (DSS-$d_6$).

NMR spectra were collected on an Avance III 700MHz NMR spectrometer (Bruker Biospin, Billerica, MA, USA) with a TXI triple resonance probe operating at 25C. A 1D NOESY pulse sequence was used with a spectral width of 12 ppm. Each spectrum was digitized with 32768 points over a 3.9 s total acquisition time. The sequence used a 100 ms mixing time and an inter scan relaxation delay of 2.0 s. These measurements were collected for only a single cohort of mice due to availability of samples.

The spectra were processed using Advanced Chemistry Development Spectrus Processor (version 2016.1, Toronto, ON, Canada). Processing parameters included zero filling to 65526 points, 0.3 Hz decaying exponential multiplication. Phase correction was applied using the automated algorithms in the ACD software and a third order polynomial baseline correction was applied to all samples. Quantitative profiling of the metabolites was carried out with the Chenomx NMR Suite (version 8.2, Edmonton, AL, Canada). The chemical shifts and concentrations were referenced using DSS-$d_6$ added to each sample at a concentration of 250 µM and set to a chemical shift of 0.00 ppm. The Chenomx 700MHz library included acetate, acetoacetate, acetoin, alanine, asparagine, aspartate, B-hydroxybutyrate, butyrate, DSS-d6, ethanol, formate, glucose, glutamine, glutamate, glycine, isoleucine, isopropanol, lactate, leucine, lysine, methionine, propionate, sarcosine, threonine, and valine. Signals in the bile acid regions of the spectrum, from approximately 0.6 to 0.9 ppm, were fit as synthetic Lorentizan functions and each signal given a generic assignment as a bile acid methyl group. Concentration for bile acids are presented as arbitrary concentration units and relative comparisons should be made. The fitting of each spectrum was carried out in batch mode. Manual adjustment of the fitting was carried out in some samples to correct for errors arising from spectral overlap.

## Ex vivo experiments

### Culture conditions and treatment

Gut microbiota were cultured from human stool in a flow-through bioreactor system (Infors, Bottmingen Switzerland) following previous work (*McDonald et al., 2013*). Healthy human subjects (n = 3) that self-reported no use of antibiotics within a month of enrollment provided a single donation of stool. The sample size was calculated using a power analysis to allow for β less than 0.1. This experiment was only performed with a single cohort of donors. Informed consent was obtained from all subjects and the protocol was approved by the Duke Health Institutional Review Board. Subjects collected samples by placing disposable commode specimen containers (Fisher Scientific, Hampton NH) under their toilet seats before bowel movements. Intact stool samples were moved within

roughly 15 min into anaerobic conditions after collection. All bioreactor vessels had a total vessel volume of 400 ml and a continuous turnover of 400 ml per day for the media to emulate the 24 hr average passage time of the human gut. The bioreactors communities were subject to a two-day adjustment period before treatment began (*Figure 2—figure supplement 1*). Vessels were then randomly assigned a treatment condition. Antibiotic treated cultures were subjected to daily single doses of 3.896 ml antibiotic treatments on two consecutive days. Antibiotics were prepared as above for mice. Control and antibiotic vessels had 1 L per minute $N_2$ gas bubbled in. Redox was measured continuously with Hamilton (Reno NV) EasyFerm Plus ORP K8 225 probes, which has a measuring range of $\pm2000$ mV. Redox probes were calibrated in Hamilton ORP buffer before autoclaving the bioreactor setup at the beginning of each replicate run. Because of differences in the construction and calibration between the in vivo and ex vivo redox probes, direct comparisons between the in vivo and ex vivo redox values should not be performed. Samples were collected daily and frozen immediately at $-80°C$ for subsequent analysis of microbial community composition. Abiotic data were averaged for five minute increments for analysis.

## Statistical analysis

We performed linear mixed effects model analysis to determine the effects of antibiotics on redox potential, oxygen concentration, fecal lipocalin-2 concentration, and Bray-Curtis dissimilarity. As fixed effects, we entered antibiotic treatment and time with an interaction term into the model. We included mouse identity as a random effect. P values were obtained by likelihood ratio tests comparing the full model against a model including only time and mouse identity and were performed with the 'anova' function in the 'lme4' package. Repeated measures correlations were used to assess correlations where multiple time points from the same mouse were included in the statistical analysis. Repeated measures correlations were calculated with the 'rmcorr' function in the 'rmcorr' package. These and all other statistical analyses were carried out in R (R core team, version 3.3). All statistical tests performed were non-parametric except where a Shapiro-Wilks test indicated that data were normally distributed, in which case parametric tests were used. All data points were included in analyses and outliers were not treated in any manner.

## Acknowledgements

Elise Cowley provided advice on experimental procedures and interpretation of redox dynamics. Lionel Watkins aided in the chemostat experiments. The manuscript was improved by comments from Cari Ficken and Robert Dunn. Inflammation assays were performed by Carlton Anderson at the UNC Center for Gastrointestinal Biology and Disease (CGIBD) Advanced Analytics Core funded by NIH grant P30-DK034987. RT-QPCR was carried out by members of the Duke Sequencing and Genomic Technologies core. This work was significantly improved by the comments of two anonymous reviewers and Jon Clardy.

## Additional information

### Competing interests

Bruce Klitzman: Duke University has received support from Profusa, Inc. as an NIH subaward and additional corporate support for research. Dr. Klitzman has no equity ownership. He now has a consulting agreement and has received professorial sponsorship through Duke University from Profusa, Inc. Scott P Nichols: Is the Director of Pre-Clinical and Advanced Technologies at Profusa, Inc. Natalie A Wisniewski: Is a Co-Founder and Chief Technology Officer of Profusa, Inc. The other authors declare that no competing interests exist.

### Funding

| Funder | Author |
| --- | --- |
| National Science Foundation | Aspen T Reese<br>Justin P Wright<br>Lawrence A David |

| Hartwell Foundation | Lawrence A David |
| Alfred P. Sloan Foundation | Lawrence A David |

The funders had no role in study design, data collection and interpretation, or the decision to submit the work for publication.

## Author contributions

Aspen T Reese, Conceptualization, Data curation, Formal analysis, Funding acquisition, Validation, Investigation, Visualization, Methodology, Writing—original draft, Writing—review and editing; Eugenia H Cho, Investigation, Methodology, Writing—review and editing; Bruce Klitzman, Methodology, Writing—review and editing; Scott P Nichols, Software, Investigation, Methodology, Writing—review and editing; Natalie A Wisniewski, Software, Methodology, Writing—review and editing; Max M Villa, Validation, Investigation, Methodology, Writing—review and editing; Heather K Durand, Sharon Jiang, Sai N Nimmagadda, Investigation, Writing—review and editing; Firas S Midani, Formal analysis, Methodology, Writing—review and editing; Thomas M O'Connell, Data curation, Software, Investigation, Methodology, Writing—review and editing; Justin P Wright, Conceptualization, Supervision, Funding acquisition, Writing—review and editing; Marc A Deshusses, Conceptualization, Writing—review and editing; Lawrence A David, Conceptualization, Formal analysis, Supervision, Funding acquisition, Project administration, Writing—review and editing

## Author ORCIDs

Aspen T Reese ⓘ https://orcid.org/0000-0001-9004-9470
Heather K Durand ⓘ https://orcid.org/0000-0003-4059-8484
Firas S Midani ⓘ https://orcid.org/0000-0002-2473-7758
Sai N Nimmagadda ⓘ https://orcid.org/0000-0003-1837-4054
Lawrence A David ⓘ https://orcid.org/0000-0002-3570-4767

## Ethics

Human subjects: Informed consent was obtained from all subjects and the protocol (Pro00049498) was approved by the Duke Health Institutional Review Board.
Animal experimentation: All animal experiments were conducted in accordance with National Institute of Health Guide for the Care and Use of Laboratory Animals using protocols approved by the Duke University Institutional Animal Care & Use Committee (Protocol A140-14-04).

## Decision letter and Author response

Decision letter https://doi.org/10.7554/eLife.35987.028
Author response https://doi.org/10.7554/eLife.35987.029

# Additional files

## Supplementary files

• Supplementary file 1. Metabolomic dynamics during and after antibiotic treatment. * indicate metabolites with overall effect of treatment (p<0.05 linear mixed effects model likelihood tests). Bolded text indicates time points with significant difference between treated and control groups (p<0.05 Bonferroni-corrected Mann-Whitney U tests). Blank cells are time points for which that metabolite was not quantified.
DOI: https://doi.org/10.7554/eLife.35987.021

• Supplementary file 2. Bacterial cluster constituent genera with taxonomy from Greengenes. Genera present in the five bacterial clusters with greater than 1% average abundance. The Typical, Staph/Strep, and Enterobacteriaceae clusters were significantly impacted during antibiotic treatment. The Enterobacteriaceae and Akkermansia clusters were significantly impacted during recovery.
DOI: https://doi.org/10.7554/eLife.35987.022

- Supplementary file 3. Host gene targets for RT-PCR. 'Sequenced' column identifies genes that were successfully quantified in a majority of mice.
DOI: https://doi.org/10.7554/eLife.35987.023

- Transparent reporting form
DOI: https://doi.org/10.7554/eLife.35987.024

## Data availability

Sequencing data have been deposited in the European Nucleotide Archive under Primary Accession Number (PRJEB26446).

The following dataset was generated:

| Author(s) | Year | Dataset title | Dataset URL | Database, license, and accessibility information |
|---|---|---|---|---|
| Aspen T Reese, Lawrence A David | 2018 | Antibiotic induced changes in the microbiota disrupt redox dynamics in the gut | https://www.ebi.ac.uk/ena/data/view/PRJEB26446 | Publicly available at the European Nucleotide Archive (accession no: PRJEB26446) |

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
