## [Decision Letter]

Thank you for submitting your article "Antibiotic induced changes in the microbiota disrupt redox dynamics in the gut" for consideration by *eLife*. Your article has been favorably evaluated by Wendy Garrett (Senior Editor) and three reviewers, one of whom is a member of our Board of Reviewing Editors. The following individual involved in review of your submission has agreed to reveal his identity: Jon Clardy (Reviewer #3).

The reviewers have discussed the reviews with one another and the Reviewing Editor has drafted this decision to help you prepare a revised submission.

Summary:

The reviewers agreed that this paper represents a very interesting study of how intestinal bacteria shape the chemical environment of the intestine. The authors address this question by characterizing changes in redox state during antibiotic treatment, finding that intestinal redox potential increases following antibiotic treatment. A major claim is that bacterial metabolic activity is the primary determinant of redox potential, and that host metabolic activity plays a lesser role. The authors also find that the antibiotic-induced changes in redox state coincide with a characteristic bloom of *Enterobacteriaceae*, which accords with numerous prior published studies showing that electron receptors, such as nitrate, support *Enterobacteriaceae* expansion.

Overall, the study represents a novel exploration of how a major perturbation to the gut microbiome, antibiotic addition, affects a key environmental factor, redox potential. The study is logical, well conducted and controlled with a straightforward experimental approach, and an impressive amount of data is included. Although some of the conclusions were unsurprising, the reviewers felt that overall, the work provides valuable insight to the gut microbiome field.

Essential revisions:

1) A major claim of this study is that the increase in redox potential after antibiotic treatment is due to bacterial metabolic activity and not because of host processes. However, the authors also show that the major electron acceptor that increases after antibiotic treatment is nitrate, which is necessarily derived from the host, and they further show that this correlates with an increase in NOS expression in the intestine. Indeed, it is not surprising that host-derived electron acceptors would increase in abundance when bacterial density is reduced, as there are fewer bacteria present to consume them. Overall, the logic underpinning their conclusion that the altered redox state in vivo is strictly due to bacterial activity needs to be clarified.

2) A question was raised about the reproducibility of the work. In the Materials and methods, it is stated that C57BL/6 mice male mice were purchased from Charles River. However, it is unclear whether the mouse work was repeated with groups of mice ordered at different times. The microbiota can differ even in mice from the same vendor, especially if mice are housed in different rooms/facilities. Thus, the authors should clarify whether the mouse microbiota was similar in mice ordered at different times.

3) Some of the statistical analyses looked marginal, which probably reflects the difficulty of making measurements on complex samples (e.g., Figure 1C). A brief explanation of what was done to the data in the text of the figure legend would help.

4) Most, if not all, the results are not surprising, and this should be made clearer. In the Abstract the last sentence is "Together, our results argue that bacterial competition for electron acceptors is a key ecological factor structuring gut microbial communities." This is known in many contexts and should be rephrased to something about a generalized function of antibiotics or implications for understanding and controlling the human microbiota and disease.

5) It would be appropriate for this manuscript to cite the 2018 Nature paper "Precision editing of the gut microbiota ameliorates colitis" by Sebastian Winter and co-workers. The Winter paper describes how inhibiting a bacterial redox process in vivo (Mo-dependent reduction) can be used to reshape the community in cases where terminal electron acceptors are determinants of community composition. It provides a nice contrast in approaches: in this manuscript a general conclusion, in the other a specific approach.

---

## [Author Response]

Essential revisions:1) A major claim of this study is that the increase in redox potential after antibiotic treatment is due to bacterial metabolic activity and not because of host processes. However, the authors also show that the major electron acceptor that increases after antibiotic treatment is nitrate, which is necessarily derived from the host, and they further show that this correlates with an increase in NOS expression in the intestine. Indeed, it is not surprising that host-derived electron acceptors would increase in abundance when bacterial density is reduced, as there are fewer bacteria present to consume them. Overall, the logic underpinning their conclusion that the altered redox state in vivo is strictly due to bacterial activity needs to be clarified.

Our intention was not to claim that altered redox state in vivo is due strictly to changes in bacterial activity. We found that redox responses were not solely due to changes in host gene expression – as evidenced by our ability to recreate in the ex vivo bioreactor system – or pathogen colonization – which had been argued elsewhere in studies which did not isolate pathogen colonization effects from antibiotic effects. Still, we agree the nitrate and NOS expression results highlight the contribution that host processes can play in shaping redox potential. As suggested, we have edited the text to clarify and balance these points:

“Although there was evidence of increases in certain host pathways of electron acceptor production, we did not find persistent and significant changes in host inflammation. Therefore, we propose microbial respiration of host sourced electron acceptors acts as a major determinant of redox potential in the gut, in addition to the host production itself…”

“But, as oxygen grew increasingly available, the remaining facultative anaerobes switched to aerobic respiration leaving other host-sourced electron acceptors, such as nitrate, to accumulate. Accrued electron acceptors, especially nitrate, may also have been contributed by changes in host physiology as evidenced by an upregulation of the *Nos2* gene, which is linked to reactive nitrogen species levels (Dedon and Tannenbaum 2004, Winterbourn 2008).”

2) A question was raised about the reproducibility of the work. In the Materials and methods, it is stated that C57BL/6 mice male mice were purchased from Charles River. However, it is unclear whether the mouse work was repeated with groups of mice ordered at different times. The microbiota can differ even in mice from the same vendor, especially if mice are housed in different rooms/facilities. Thus, the authors should clarify whether the mouse microbiota was similar in mice ordered at different times.

This is an important question given the variation in gut microbiota known to exist between mice both within and between vendors (Rosshart et al. 2017, Rausch et al. 2016, Ivanov et al. 2009, Campbell et al. 2012). Indeed, our mouse experiments were repeated with different groups of mice ordered 4 months apart, although they were housed in the same room at the vivarium during each experimental period.

In response to overall concerns about reproducibility, we re-analyzed the two cohorts of mice from which we collected 16S rRNA gene sequence data (see Figure 4—figure supplement 5). We observed significant differences in community composition at the OTU level based on mouse cohort, which appeared to persist throughout the experiment (P=0.01, R^2^=0.17; PERMANOVA). This suggested that despite coming from the same vendor, our mice indeed had different microbiota. Still, significant compositional changes accompanied antibiotic treatment (P=0.01, R^2^=0.07; PERMANOVA). These treatment effects were not dependent on initial community composition (i.e. there was not a significant interaction between cohort and treatment (P=0.3)). Moreover, we note that the overall bloom of *Enterobacteriaceae* in response to treatment has been shown by other groups in using different mouse models and in multiple animals under many antibiotic types; and, the comparability in compositional shifts and environmental shifts documented in our human ex vivo gut model. These findings support the replicability of high level features of our mouse results.

Nevertheless, we appreciate overall concerns regarding the reproducibility of mouse work and now discuss both our re-analysis, as well as limitations regarding analysis of mouse litters from different time points:

“Because baseline microbiome is expected to differ between mouse litters over time as well as between vendor (Rosshart et al. 2017, Rausch et al. 2016, Ivanov et al. 2009, Campbell et al. 2012), we examined whether the effects observed were reproducible between the two cohorts sequenced. […] Because the *Enterobacteriaceae* bloom response to antibiotics has been found to be consistent between humans and mice, as well as between mouse vendors (Antonopoulos et al. 2009, David et al. 2015, Young and Schmidt 2004, Theriot et al. 2014, Peterfreund et al. 2012, Dethlefsen et al. 2008, Jakobsson et al. 2010, Looft and Allen 2012), we believe that the overall trends observed here are indicative of what may be occurring in other conventional gut microbiota contexts.”

3) Some of the statistical analyses looked marginal, which probably reflects the difficulty of making measurements on complex samples (e.g., Figure 1C). A brief explanation of what was done to the data in the text of the figure legend would help.

Two main statistical test classes were used in this paper when analyzing the time series of biotic and abiotic variables. To assess overall impacts of treatment, we ran linear mixed effects models and tested for increased model performance in models including treatment as a fixed effects versus models which only included time as a fixed effect (in all models mouse ID was included as a random effect to correct for inter individual differences). These were the results reported in the text and the figure legends. In addition, we performed non-parametric post-hoc tests (Mann-Whitney U tests) to determine at which time points the treatment groups differed. These post-hoc tests were indicated on the figures to allow for ease of interpretation, but they were not considered the optimal assessment of treatment effects because they do not take into account trends in individuals over time or repeated measure non-independence, and they require extensive multiple-hypothesis correcting. As such, results such as the lipocalin-2 occurred wherein there was an overall effect of treatment captured in the mixed effects models but only small differences between treatment groups at any given time point. Due to the limitations of using post-hoc tests to analyze time series, the non-parametric single time point results were not included in the main text or specified in the legends.

Still, we acknowledge how our decisions regarding data presentation gave the appearance that some statistical outcomes were marginal. To clarify the differences between test types, and direct readers attention mainly to the mixed effects model results, we have adjusted all figure legends to read “Post-hoc test results (* indicates P<0.05 Bonferroni-corrected Mann-Whitney U test) are included for visualization purposes” where previously they read “* indicates time points where post-hoc tests indicate significant difference from control (P<0.05 Bonferroni-corrected Mann-Whitney U test)”. In addition we have updated all figures to include the statistical results for the mixed effects model test on any panel where such a test is being reported.

4) Most, if not all, the results are not surprising, and this should be made clearer. In the Abstract the last sentence is "Together, our results argue that bacterial competition for electron acceptors is a key ecological factor structuring gut microbial communities." This is known in many contexts and should be rephrased to something about a generalized function of antibiotics or implications for understanding and controlling the human microbiota and disease.

We believe it notable that many of our results had not been previously shown in assessments of the effects of antibiotics. Furthermore, while individual electron acceptors are sometimes considered in studies of the gut microbiota, they are rarely presented in concert nor is overall redox potential measured. Nevertheless, we do acknowledge that many of our results could have been predicted from first principles. Hence, we have now tempered the language in the abstract and refocused the final sentence for implications for understanding the relationship between the gut microbiota and health. The new sentence is reproduced below:

“As bacterial competition for electron acceptors can be a key ecological factor structuring microbial communities, these results support the potential for manipulating gut microbiota through managing bacterial respiration.”5) It would be appropriate for this manuscript to cite the 2018 Nature paper "Precision editing of the gut microbiota ameliorates colitis" by Sebastian Winter and co-workers. The Winter paper describes how inhibiting a bacterial redox process in vivo (Mo-dependent reduction) can be used to reshape the community in cases where terminal electron acceptors are determinants of community composition. It provides a nice contrast in approaches: in this manuscript a general conclusion, in the other a specific approach.

We appreciate you calling our attention to this paper. It is now discussed and cited:

“*Enterobacteriaceae* and respiration pathway genes more generally are also more prevalent in inflamed mouse guts (Hughes et al. 2017), another environment where heightened redox potential is expected. […] This result corroborates our model that members of the *Enterobacteriaceae* use respiration to bloom under disturbed conditions, and it highlights the potential for manipulating electron acceptor availability or usability to engineer the microbiota.”